# C-REV Retains High Infectivity Regardless of the Expression Levels of cGAS and STING in Cultured Pancreatic Cancer Cells

**DOI:** 10.3390/cells10061502

**Published:** 2021-06-15

**Authors:** Daishi Morimoto, Shigeru Matsumura, Itzel Bustos-Villalobos, Patricia Angela Sibal, Toru Ichinose, Yoshinori Naoe, Ibrahim Ragab Eissa, Mohamed Abdelmoneim, Nobuaki Mukoyama, Noriyuki Miyajima, Maki Tanaka, Yasuhiro Kodera, Hideki Kasuya

**Affiliations:** 1Department of Surgery II, Graduate School of Medicine, Nagoya University, Nagoya 466-8550, Japan; d-morimoto@med.nagoya-u.ac.jp (D.M.); brahimessa@med.nagoya-u.ac.jp (I.R.E.); mohamedali@med.nagoya-u.ac.jp (M.A.); ykodera@med.nagoya-u.ac.jp (Y.K.); 2Cancer Immune Therapy Research Center, Graduate School of Medicine, Nagoya University, Nagoya 466-8550, Japan; smatsumu@med.nagoya-u.ac.jp (S.M.); bustositzel@med.nagoya-u.ac.jp (I.B.-V.); ninia.sibal@gmail.com (P.A.S.); tichinose@bwh.harvard.edu (T.I.); naoepenn@gmail.com (Y.N.); 3Department of Biological Science, School of Science, Nagoya University, Nagoya 466-8550, Japan; 4Faculty of Science, Tanta University, Tanta 31111, Egypt; 5Department of Microbiology, Faculty of Veterinary Medicine, Zagazig University, Zagazig 44519, Egypt; 6Department of Otolaryngology, Graduate School of Medicine, Nagoya University, Nagoya 466-8550, Japan; mkym1117@med.nagoya-u.ac.jp; 7Department of Transplantation and Endocrine Surgery, Graduate School of Medicine, Nagoya University, Nagoya 466-8550, Japan; miyajima@med.nagoya-u.ac.jp; 8Takara Bio Inc., Kusatsu, Shiga 525-0058, Japan; tanakamy@takara-bio.co.jp

**Keywords:** oncolytic virus, cGAS–STING pathway, human pancreatic cancer cell lines

## Abstract

Oncolytic virus (OV) therapy is widely considered as a major breakthrough in anti-cancer treatments. In our previous study, the efficacy and safety of using C-REV for anti-cancer therapy in patients during stage I clinical trial was reported. The stimulator of interferon genes (STING)–TBK1–IRF3–IFN pathway is known to act as the central cellular host defense against viral infection. Recent reports have linked low expression levels of cGAS and STING in cancer cells to poor prognosis among patients. Moreover, downregulation of cGAS and STING has been linked to higher susceptibility to OV infection among several cancer cell lines. In this paper, we show that there is little correlation between levels of cGAS/STING expression and susceptibility to C-REV among human pancreatic cancer cell lines. Despite having a responsive STING pathway, BxPC-3 cells are highly susceptible to C-REV infection. Upon pre-activation of the STING pathway, BxPc-3 cells exhibited resistance to C-REV infection. However, without pre-activation, C-REV completely suppressed the STING pathway in BxPC-3 cells. Additionally, despite harboring defects in the STING pathway, other high-grade cancer cell lines, such as Capan-2, PANC-1 and MiaPaCa-2, still exhibited low susceptibility to C-REV infection. Furthermore, overexpression of STING in MiaPaCa-2 cells altered susceptibility to a limited extent. Taken together, our data suggest that the cGAS–STING pathway plays a minor role in the susceptibility of pancreatic cancer cell lines to C-REV infection.

## 1. Introduction

Pancreatic cancer has been considered as one of the major causes of cancer-related mortalities for several years, with a 5-year survival rate of only 8% worldwide [1,2]. Pancreatic ductal adenocarcinoma (PDAC) accounts for 95% of all pancreatic cancers. Although several advancements in operative procedures have been developed, less than 20% of patients have tumors that are surgically resectable [2]. Of these, 80% are estimated to relapse after surgery [3,4]. Although recent cancer therapy advancements, especially immune checkpoint inhibitors, were found to significantly increase prognostic outcomes in other types of solid cancers, only marginal benefits were reported for PDAC [5,6,7]. In some cases, no clinical response was observed among PDAC patients during the phase I clinical trial of an anti-PD-L1 therapy [7,8]. Given the current limited progress in therapy methods, pancreatic cancer is expected to rise from being the fourth to the second leading cause of cancer-related mortality within the next decade [2].

Several studies have attributed the high resistance of pancreatic cancer to immunotherapies to its distinct tumor microenvironment [9]. Compared to other types of solid tumors, PDAC has a more immunosuppressive tumor microenvironment. A key factor in maintaining the PDAC microenvironment is its dense desmoplastic stromal reaction [9]. This desmoplastic reaction is characterized by a strong crosstalk between tumor cells and stromal fibroblasts that facilitates the formation of solid stiff tumor tissues [10]. This high solid stiffness establishes a barrier that drastically limits the infiltration of effector T-cells, unlike in other types of solid tumors, in which effector-T cell infiltration is prominent [11]. To add to this, the overexpression of immune checkpoint signaling molecules by the tumors and a high number of suppressive macrophages, dendritic cells and regulatory T cells present in the tumor microenvironment exhaust the already limited amount of CD8T cells [11]. PDAC has also been shown to express low levels of MHC class I, which allows tumor cells to evade the attack by CD8T cells. Indeed, it has been suggested that MHC class I-expressing PDAC are eliminated by tumor specific CD8T cells and the remaining cells maintain MHC class I negative status by activation of the ER stress pathway [12].

Through the years, oncolytic viral therapy has been seen as a major breakthrough therapy that has the potential to overcome the limitations posed by the immunosuppressive tumor microenvironment of PDAC. Because oncolytic virus entry depends only on viral entry receptors, the oncolytic virus is capable of infecting tumor cells with low MHC class I presentation. More than 30 oncolytic viruses (OVs) have been conducted from preclinical research to clinical research on various cancers, including pancreatic cancer [13,14,15]. The first oncolytic virus to be granted approval by the FDA in several countries is a Herpes Simplex Virus Type-1 based oncolytic virus (OVs), talimogene laherparepvec (T-VEC, ImlygicTM, Amgen, Cambridge, UK), armed with granulocyte-macrophage colony-stimulating factor (GM-CSF) [16,17,18]. Aside from T-VEC, other HSV-1 based OVs have already been garnering clinical relevance. C-REV (formerly HF-10) is a spontaneously occurring herpes simplex virus type 1 (HSV-1) mutant that is lacking functional expression of UL43, UL49.5, UL55 and UL56 [19,20]. Our laboratory has previously reported the efficacy and safety for patients of C-REV with unresectable pancreatic cancer in phase I dose-escalation clinical trials. Clinical application of C-REV is further advanced in melanoma and combination therapy of C-REV and ipilimumab (NCT02272855) has already completed patient enrollment. Despite differences among different OVs, preclinical studies have shown that all OV treatments generate an acute strongly inflamed environment that can overcome the immunosuppressive tumor microenvironment, thereby allowing it to be easily recognizable by the immune system [13,21]. Although OV treatments hold great potential, their efficacy depends on several key factors, including their replication efficiency in the tumor cells.

In general, the highly proliferative status of tumor cells facilitates rapid replication of OVs, thus making OVs specifically replication competent in tumor cells [13]. In addition, tumor cells have developed major defects in the apoptotic pathway and Protein kinase R (PKR) pathway, which play a huge role in the anti-viral defense mechanisms of the cell [22]. Furthermore, it has been reported that advanced tumor cells have developed defects in the STING pathway to suppress type I IFN induction [23,24]. Because induction of type I IFN has a central role for anti-viral immune defense, advanced tumor cells would provide a more ideal environment to viruses [23,24]. The STING pathway has several sensor molecules for detecting infectious enemies [25,26]. cGAS functions as a sensor for short double strand DNA (dsDNA), especially viral DNA of invading pathogens, such as HSV-1 [25,26]. In tumor cells, cGAS also detects self-DNA fragments leaked from the nucleus contributing to immune rejection [27,28]. Upon association of cGAS with dsDNA, cGAS is activated, which then allows it to catalyze the synthesis of cyclic GMP-AMP (cGAMP) from ATP and GTP, which, in turn, leads to STING activation on the ER-Golgi intermediate compartment and the Golgi apparatus [25,26,29]. Subsequently, TBK1, activated by STING, phosphorylates IRF3 to form IRF3 dimers [25,26]. The dimerized phosphorylated IRF3s translocate into the nucleus to start rapid transcription of type I IFNs [25,26]. Therefore, more advanced tumor cells with defects in the STING pathway provide seemingly ideal environments for oncolytic virus replications. This hypothesis has been challenged in several types of tumors, including human colon adenocarcinoma and human melanoma, showing that loss of the STING function contributed to enhanced susceptibility to oncolytic HSV-1 [23,24]. However, HSV1 has several viral genes which counteract against the cGAS–STING–IFN axis [30]. Indeed, VSV-oncolytic virus showed higher sensitivity against type-I IFN cellular responses than HSV-1, or vaccinia virus [31]. Since ICP34.5 has multiple ways to overcome cellular defenses, including IFN responses, we hypothesized that C-REV, which has intact ICP 34.5 expression, might be mildly sensitive to the cGAS–STING–IFN axis.

In this paper, we presented that with use of several human pancreatic adenocarcinoma cell lines, cell lines with defects in the STING pathway still have relatively low susceptibility to C-REV, while the other cell lines with a responsive STING pathway have relatively high susceptibility to C-REV. In both types, STING pathway activation conferred partial resistance to C-REV, showing that the STING pathway does affect replication of OVs; however, it is not a major contributor to C-REV in human PDAC cell lines.

## 2. Materials and Methods

### 2.1. Cell Lines

Human pancreatic cancer cell lines BxPC-3, AsPC1/CMV-Luc, Capan-2, PANC-1 and MiaPaCa-2 cells were cultured in Dulbecco’s modified eagle medium with high glucose (DMEM; Wako, Japan) and supplemented with 10% heat-inactivated fetal bovine serum (FBS; Biosera, France), 100 IU/mL penicillin and 100 μg/mL streptomycin (Wako) at 37 °C, in an incubator containing 5% CO_2_ humidified atmosphere. African green monkey kidney (Vero) cells were used for virus plaque assays and for preparation of virus stocks. Cell lines stably over expressing human STING were generated by integration of linearized plasmids carrying the Hygromycin resistance cassette and isolated by picking colonies.

### 2.2. Viruses

C-REV is an attenuated mutant clone derived from the HSV-1 strain HF. The virus was propagated in Vero cells and stored in aliquots at −80 °C. C-REV was diluted in DMEM without FBS for experiments. Viral titers were assayed in Vero cells and are expressed as plaque-forming units per milliliter (PFU/mL). C-REV-GFP were generated by homologous recombination with target vector containing homologous arms in the UL42–UL43 region with a CMV-IRES-ZsGreen-pA cassette.

### 2.3. Cell Proliferation Assay

Cell proliferation was determined using the 3-(4,5-dimethylthiazol-2-yl)-2,5-diphenyl tetrazolium bromide (MTT) dye reduction method. Cells were seeded, grown in DMEM containing 5% FBS and incubated at 37 °C with 5% CO_2_ overnight. Cells were infected with C-REV at several multiplicities of infection (MOIs) for 1 h in DMEM without FBS at 37 °C with 5% CO_2_. After 1 h incubation, the culture medium was adjusted to 5% FBS in final concentration. Viable cells were then quantified by colorimetric MTT assay on the second and third day after infection.

### 2.4. 2′3′-cGAMP Treatment

Cells were seeded in DMEM with 10% FBS. The medium was then replaced with a buffer containing 50 mM Hepes (pH 7.4), 100 mM KCl, 3 mM MgCl_2_, 10 μg/mL digitonin, 0.1 mM DTT, 85 mM Sucrose, 0.2% BSA, 1 mM ATP and 2′3′-cGAMP. Cells were incubated for 30 min at 37 °C with 5% CO_2_, after which, the buffer was replaced with DMEM medium and incubated for 30 min at 37 °C with 5% CO_2_. After incubation, cells were infected with C-REV as described.

### 2.5. Western Blotting

Cells were lysed in 1× sample buffer (20 mM Hepes pH 7.3, 25 mM 2-glycerophosphate, 50 mM NaCl, 1.5 mM MgCl_2_, 2 mM EDTA, 2% SDS, 5% β-ME) and boiled for 10 min. The cell lysate was then subjected to immunoblotting with anti-phospho-TBK1 (CST, #5483), anti-TBK1 (CST, #3504), anti-phospho-IRF3 (CST, #4947), anti-IRF3 (CST, #10949), anti-STING (CST, #13647), anti-cGAS (#15102, Cell Signaling Technology Japan, Tokyo, Japan), anti-α-tubulin (sc-5286) and anti-β-actin (sc-81178, Santa Cruz Biotechnology, Inc., Santa Cruz, CA, USA) antibodies.

### 2.6. Cell Staining

For the detection of STING and IRF3, cells were fixed with 4% paraformaldehyde, followed by incubation with 0.5% Triton X-100 in PBS at RT for 10 min. Cells were then blocked with 3% BSA at RT for 30 min, incubated with primary antibodies at 4 °C overnight, washed and incubated for 1 h with secondary antibodies (Rhodamine Red-X-conjugated AffiniPure F(ab’) Fragment Donkey anti-mouse or anti-rabbit; Jackson Immuno Research Labs. Inc., West Grove, PA, USA). Images were obtained with the BZ-X800 microscope (KEYENCE CORPORATION, Osaka, Japan).

### 2.7. Quantitative Real-Time PCR (qRT-PCR)

The total RNA (0.5 μg) was isolated with the Trizol reagent (Invitrogen) and reverse transcribed to generate cDNA using ReverTra Ace (TOYOBO) for 1 h at 42 °C. The resulting cDNA was used as a template for qRT-PCR quantification of IFN β using THUNDERBIRD SYBR qPCR Mix (TOYOBO). The primer sequences used in this study are below. Human GAPDH fw: 5′-CCTGCACCACCAACTGCTT-3′; human GAPDH rev: 5′-GGCCATCCACAGTCTTCTGAG-3′; human IFN-β fw: 5′-GGAGGACGCCGCATTGAC-3′; human IFN-β rev: 5′-TGATAGACATTAGCCAGGAGGTTC-3′. Quantification was carried out on a StepOne Real-time PCR system (Thermo Fisher Scientific, Waltham, MA, USA) for cDNA amplification under the following conditions: 95 °C for 10 min, followed by 40 cycles of 95 °C for 30 s and 60 °C for 1 min. Relative mRNA levels were determined using the comparative CT method and normalized against GAPDH mRNA.

### 2.8. Statistical Analysis

Statistical comparisons were performed using the Prism software, version 7.05 (GraphPad Software, San Diego, CA, USA). Statistical significance between two groups was analyzed using unpaired Student’s *t*-test. Among more than three groups, two-way ANOVA tests–Dunnett’s multiple comparison tests were performed to calculate *p*-values. *p*-values < 0.05 were considered to be statistically significant.

## 3. Results

### 3.1. Susceptibility to C-REV Varies among Human Pancreatic Cancer Cell Lines

The oncolytic virus has a huge potential to evoke strong antitumor immune responses; however, its efficacy is highly dependent on its cytotoxic effects and its replication efficiency on tumor cells. To evaluate the cytotoxic effects of C-REV on several types of human pancreatic adenocarcinoma cell lines, cells were infected with C-REV at several multiplicity of infection (MOI) and the cell viability was measured by MTT assay on day 2 and day 3 (Figure 1A). AsPC-1 and BxPC-3 cells depicted high susceptibility to C-REV, as these cells were drastically eliminated even at a relatively low MOI of 0.1 and 0.01 on day 3. In contrast, PANC-1, MiaPaCa-2 and Capan-2 cells showed low susceptibility to C-REV. It is probable that such differences in susceptibility can be attributed to different degrees of differentiation among the cell lines. Several studies, however, have reported that, except for Capan-2 cells, all the other cell lines have an undifferentiated status, a key feature of advanced PDAC tumor cells [32,33]. Several studies have also proposed that a common feature among advanced tumor cells is a highly defective STING pathway [23,24]. From this, we hypothesized that the different susceptibility to C-REV infection among cell lines can be attributed to the different degrees of defects in the STING pathway among cell lines (Figure 1B). To estimate the intactness of the STING pathway, STING and cGAS protein expressions of each cell line were examined (Figure 1C). AsPC1, BxPc-3 and Capan-2 expressed relatively abundant STING proteins but did not express cGAS highly. In contrast, PANC-1 and MiaPaCa-2 showed relatively high expression of cGAS but did not exhibit STING expression. Based on the expression pattern of these two STING components, all the cell lines seemed to be defective in the STING pathway, although in varying degrees. Next, to test the possibility that cells have different degrees of permissiveness for viral entry, cells were infected with C-REV-GFP viruses at MOI of 5. After 6 h post infection (hpi), except for Capan-2, all cell lines displayed similar percentages of GFP positive cells. Similarly, at 20 hpi, most of the cells were already GFP positive in all the cell lines, but only a few cells were GFP positive in Capan-2 (Figure 1D).

### 3.2. C-REV Susceptible BxPC-3 Cell Line Has a Functional STING Pathway

Since BxPC-3 cells displayed the highest susceptibility to C-REV, we initially speculated that the cells might be completely unable to activate the STING pathway. Because BxPC-3 cells expressed small amounts of cGAS, we stimulated the cells with short double stranded DNA (90 nucleotides) (dsDNA90) by transfection. Six hours after transfection of DNA, induction of IFNβ was measured by RT-qPCR. We found that cells were able to induce IFNβ mRNA (Figure 2A). To test if BxPC-3 cells were able to activate the STING pathway, the phosphorylation statuses of TBK1 and IRF3 were examined by western blot analysis, which showed activation of the STING pathway in both dsDNA90 and PolyI:C stimulation (Figure 2B). To confirm this, the cells were tested by immunofluorescent staining of STING and IRF3. Consistent with previous reports, upon stimulation with dsDNA90, STING was stacked with Golgi-like structures and IRF3 was translocated into the nucleus (Figure 2C, arrowheads). These results suggest that BxPC-3 cells are able to activate the STING pathway even though the expression level of cGAS proteins is low. Because STING expression levels in many types of cancer cells are relatively low compared to those of normal cells, we thought that BxPC-3 cells can fully activate the STING pathway if STING is over expressed. Thus, we made STING over-expressing cells and examined the activation status of the STING pathway (BxPC-3 STING o/e #8: ST-8 and #10: ST-10). Contrary to our initial assumption, we found that increasing the level of STING expression showed little difference in the activation status (Figure 2A (ST-8 and ST-10), 2B (ST-8 and ST-10) and 2C (ST-8 arrow)).

### 3.3. C-REV Suppresses STING Activation in BxPC-3 Cell Line

To evaluate whether the pre-activation of the STING pathway in BxPC-3 cells can suppress susceptibility to C-REV infection, cells were treated with 2′3′-cGAMP at 200 nM and 5 μM 1 h before viral infection. MTT assay results showed that BxPC-3 cells treated with a high dose of 2′3′-cGAMP displayed low susceptibility in relatively low MOI (MOI of 0.01 and 0.1); however, little difference was observed at MOI of 1 between the treated and non-treated BxPC-3 cells (Figure 3A, BxPC-3). To further see whether overexpression of STING is sufficient to suppress C-REV infection, we generated two BxPC-3 cell lines overexpressing STING and performed the same 2′3′-cGAMP treatment. Contrary to the results shown by parent BxPC-3 cell lines, STING overexpressing BxPC-3 cell lines displayed lower susceptibility to C-REV than that of BxPC-3 cells in relatively low MOI (MOI of 0.01 and 0.1), even without treatment of 2′3′-cGAMP (Figure 3A, BxPC-3 ST-8, ST-10, mock). Altogether, it was observed that a low dose of 2′3′-cGAMP treatment resulted in little difference in susceptibility, compared to mock treatment; however, a high dose of 2′3′-cGAMP treatment resulted in much lower susceptibility at MOI of 0.01 and 0.1 (Figure 3A, BxPC-3 ST-8, ST-10, cGAMP 5 μM). Interestingly, at MOI of 1, these differences in susceptibility disappeared. Activation of the STING pathway in the 2′3′-cGAMP treated cell lines was confirmed by western blotting (Figure 3B). With the 2′3′-cGAMP treatment, the amount of phosphorylated IRF3 increased slightly. These results suggest that pre-activation of the STING pathway decreases susceptibility to C-REV. Moreover, an increase in the amount of the STING protein resulted to a STING pathway that was more responsive to viral infection. Interestingly, despite this increase in sensitivity, most of the cells were still killed by virus at MOI of 1. From this, we examined whether C-REV has an ability to suppress the activation of the STING pathway without pre-activation of the STING pathway. At 6 hpi, BxPC-3 cells were stained with IRF3 and STING. At MOI of 5, most of cells were infected by virus and only a few cells showed translocation of IRF3 into nucleus and stacking of STING, suggesting that the STING pathway was not activated (Figure 3C, arrowhead). Additionally, western blot analysis showed suppression of STING pathway activation (Figure 3D). Indeed, IFNβ induction was clearly suppressed by the viral infection, compared to IFNβ induction by dsDNA90 stimulation (Figure 2A and Figure 3E). Taken together, C-REV infection with relatively high MOI resulted in complete suppression of STING pathway activation in BxPC-3 cells, or STING overexpressing BxPC-3 cells, despite having a responsive STING pathway.

### 3.4. Low Susceptibility Is Not Highly Dependent on STING Pathway Activation

Next, we examined the STING pathway in lowly susceptible cell lines, namely Capan-2, PANC-1 and MiaPaCa-2 cells. Capan-2 showed the lowest susceptibility toward C-REV. Thus, we examined viral infectivity using flow analysis on the cells infected with C-REV-GFP at 12 hpi (Figure 4A, left). AsPC1 cells showed that more than 80% of cells were expressing GFP at MOI of 3, 10 and 30, suggesting that AsPC cells were highly susceptible to the virus, which is consistent with the results in Figure 1A. On the other hand, Capan-2 cells showed the lowest percentages for infected GFP+ cells, similar to the results in PANC-1 cells. Interestingly, an increasing of MOI by 10 times had a limited effect on the increasing percentages of infected cells by 2-fold (Figure 4A, right). To know more accurate kinetics in the viral infectivity toward Capan-2 cells, we examined the cytotoxicity of C-REV on day 4 and day 5 (Figure 4B), suggesting that Capan-2 has the lowest susceptibility among five cell lines but cells are still able to be infected and killed by C-REV. Since the expression patterns of STING and cGAS were similar to those of AsPC1 cells and BxPC-3 cells, we wondered if the activation statuses of the STING pathway were also similar. Therefore, we examined the induction of IFNβ after transfection of dsDNA90 or after infection of C-REV (Figure 4C). However, the induction of IFNβ was little, compared to the successful induction in BxPC-3 cells, suggesting that Capan-2 cells have a defective STING pathway. To test this, Capan-2 cells were examined for STING subcellular localization. Surprisingly, some cells displayed stacked STING localization, implying the activation of STING in both dsDNA90 treated cells and viral infected cells (Figure 4D, arrowhead). To evaluate whether the pre-activation of the STING pathway can suppress susceptibility to C-REV infection, Capan-2 cells were treated with 2′3′-cGAMP at 5 μM 1 h, before viral infection, followed by MTT assay. As expected, little difference was observed between cells treated with or without 2′3′-cGAMP (Figure 4E). These results suggest that STING activation is not able to lead to the induction of IFNβ in Capan-2 cells. To test the downstream of the STING pathway, the phosphorylation statuses of TBK1 and IRF3 were examined by western blot analysis, which showed a slightly increasing amount of phosphorylated TBK1, while no detectable phosphorylation of IRF3 was observed (Figure 4F). Taken together, these results suggest that despite possessing a defective STING pathway, Capan-2 cells have low susceptibility toward C-REV infection.

Lastly, we examined PANC-1 cells and MiaPaCa-2 cells. Since these two cell lines express limited STING proteins, we speculated that their STING pathway would be defective. To test this, the induction of IFNβ was analyzed in both cell lines upon addition of dsDNA90. As expected, little induction was observed (Figure 5A), compared to a clear induction in BxPC3 cells (Figure 2A). In addition, immunofluorescent staining of IRF3 showed that stimulation with either dsDNA90 or PolyI:C had little effect on IRF3 localization (Figure 5B). Finally, the lack of phosphorylated IRF3 in the western blotting results confirmed the defects in the STING pathway in these two cell lines (Figure 5C,G). Because MiaPaCa-2 cells exhibited notable cGAS expression, despite their lack of STING expression, we assumed that overexpression of STING might restore the STING pathway. Thus, we generated two cell lines; one of the two showed a moderate expression of STING, while the other showed an intensive expression of STING (Figure 5C). Using these cell lines, we examined the activation status of the STING pathway after stimulation with dsDNA90 or PolyI:C (Figure 5C). Compared to a clear induction in BxPC3 cells (Figure 2A), interestingly, only cells with an intensive expression of STING were able to activate the STING pathway (Figure 5C). Because dsDNA90 or PolyI:C were introduced into cells by transfection, the results might just reflect the low transfection efficiency toward the MiaPaCa-2 cells. To check this possibility, 2′3′-cGAMP treated and nontreated cells were examined by MTT assay. Little difference was observed between PANC-1 cells treated with or without 2′3′-cGAMP (Figure 5D). Similarly, little difference was observed among parental MiaPaCa-2 cells with mock treatment, MiaPaCa-2 STING over expressing cells with mock treatment and cells with 2′3′-cGAMP treatment (Figure 5E). These data suggest that susceptibility to C-REV in PANC-1 cells or MiaPaCa-2 cells is not highly dependent on STING pathway activation. Finally, we examined the effects of C-REV infection on the STING pathway in PANC-1 cells and MiaPaCa-2 cells. C-REV infection did not induce IFNβ expression (Figure 5F). Accordingly, no activation of the STING pathway was detected in PANC-1 cells nor MiaPaCa-2 cells (Figure 5G,H). Although MiaPaCa-2 cells with over expression of STING showed an increasing amount of phosphorylated TBK1, no phosphorylation of IRF3 was observed (Figure 5H). Taken together, our results suggest that C-REV infection had little effect on IFNβ induction via STING pathway activation in PANC-1 cells and MiaPaCa-2 cells.

## 4. Discussion

In this study, we demonstrated that susceptibility to C-REV in vitro is not affected gravely by the state of the STING pathway in human pancreatic cancer cell lines. As reported previously, AsPC-1, BxPC-3, PANC-1 and MiaPaCa-2 possess undifferentiated features [32]. There are several papers showing that advanced tumor cells become defective in the STING pathway to evade from the immune system [23,24], by avoiding the secretion of type I IFNs, which alert the immune system. Consistent with this hypothesis, all these five cell lines had limited expression levels of STING or cGAS (Figure 1B).

In BxPC-3 cells, we were able to detect the activation of the STING pathway even with low expression of cGAS (Figure 2). Indeed, BxPC-3 cells showed quick activation of the STING pathway by sensing dsDNAs or 2′3′-cGAMP, which resulted in INFβ induction (Figure 2A). However, it has been reported that STING can be activated by dsDNA via direct binding to STING [34]. Thus, there is a possibility that dsDNAs stimulated STING directly in BxPC-3 cells. Similarly, considering their low expression of cGAS, Capan-2 cells might also be able to sense dsDNA via STING molecules (Figure 4D,F). Although STING molecules were activated in Capan-2 cells, our western blott analysis showed that only phosphorylation of TBK1 was detected, but no phosphorylation of IRF3 was observed (new Figure 4F). Note that a similar pattern was observed from PANC-1 cells and MiaPaCa-2 cells (Figure 5C,G). From these, we speculate that, similarly to PANC-1 and MiaPaCa-2 cells, phosphorylation of IRF3 in Capan-2 cells might be dephosphorylated by protein phosphatase-1, which is reported as a negative regulator of IRF3 [35].

Contrary to BxPC-3 cells, Capan-2 cells, PANC-1 cells and MiaPaCa-2 cells exhibited more severe defects in the STING pathway (Figure 4 and Figure 5). Nevertheless, all these three cell lines showed relatively low susceptibility to C-REV (Figure 1A). Therefore, we speculate that the resistance of these three cell lines to C-REV infection can be attributed to different factors. In fact, existing studies have shown that, compared to BxPC-3 cells and AsPC-1 cells, PANC-1 and MiaPaCa-2 exhibit features for epithelial to mesenchymal transition, including loss of expression in E-cadherin and upregulation of vimentin [36]. Consistently, PANC-1 cells and MiaPaCa-2 do not form cell-islands on a culture dish, while BxPC-3 cells and AsPC-1 cells do. Because C-REV has a feature to give a fusion phenotype, cell–cell adhesion would affect the way of viral spreading. From this, it can be deduced that cellular behaviors could be a determining factor for susceptibility in vitro.

It is important to note that viral factors can be a contributing factor to susceptibility. HSV1 has a strategy to carry its genome to the nucleus directly by using its capsid to evade viral genome detection by cGAS in the cytoplasm. Previous studies have reported that IFI16 acts as a sensor for dsDNA in the nucleus [37], thereby also making it a valid sensor for HSV1 viral infection. Taking our results together, only Capan-2 cells were able to detect HSV1 viral infection and, in turn, activate STING molecules. Considering that Capan-2 cells have low expression of cGAS, these results imply that Capan-2 cells have intact IFI16. Although cells have various viral genome sensors, which can lead to the activation of the STING molecule, C-REV contains a duplicate of the ICP27 gene, which can inhibit phosphorylation of IRF3 [38,39]. Thus, we speculate that C-REV might have an efficient inhibitory effect against the STING pathway through its high expression levels of ICP27. Additionally, it has been reported that HSV-1 is not significantly sensitive to the IFN–Jak–STAT pathway, compared to other viruses, such as oncolytic VSV [31], because HSV-1 has several viral genes to suppress the pathway for IFN induction and IFN response [30]. Consistently, our results showed that C-REV was able to suppress IFN induction completely in BxPC-3 cells (Figure 3C,E). Nevertheless, C-REV has been established as an oncolytic virus, indicating that C-REV is not able to replicate inside normal cells. Our results have shown that STING overexpressing BxPC-3 cells were relatively highly resistant upon pre-activation of the STING pathway (Figure 3A), indicating that IFNs functions as anti-C-REV infection. Despite this high resistance, C-REV overcame this resistance with a dose of MOI of 1. Likewise, the ability of HSV-1 to suppress IFN induction could also be a factor in the lack of IFN induction observed in Capan-2 cells upon C-REV infection. Although Capan-2 cells showed the least susceptibility, C-REV managed to overcome this resistance with a dose of MOI of 10 and 20. Compared to non-tumor cells, the low susceptibility in the Capan-2 cell line is not sufficient to completely reject the C-REV infection. With this, it is highly probable that in non-tumor cells, STING pathway activation would induce IFNs more intensely than BxPC-3 cells and would be far more responsive than Capan-2 cells, sufficient to suppress C-REV replication.

In this paper, we mainly focused on cellular response against C-REV oncolytic virus in human pancreatic cancer cell lines in vitro. To better elucidate the role of the cellular defense system, specifically the status of STING pathway activation on cancer cells against C-REV infection, we tried to separate the factors from the host’s immune system by utilizing the in vitro system. Interestingly, previous reports have shown that cGAMP can be incorporated into the viral particles and affect the responses in the neighbor cells and immune cells in the next round of infection [40]. Therefore, the level of cGAS activation would be a factor for activation of immune cells in vivo. Since several factors at play in C-REV infection in vivo are not adequately represented in vitro, including the tumor microenvironment, a study evaluating the effects of such factors in C-REV susceptibility should be explored in the near future. Moreover, other reports have shown that, in other cancer cell lines, an HSV1-based oncolytic virus has shown strong infectivity to the cells with a defective STING pathway [23,24]. Because this study only focused on five human pancreatic cancer cell lines and C-REV, it would be interesting to evaluate the effect of the intactness of the STING pathway against the infectivity of other HSV-1 based oncolytic viruses in other cell lines in future studies.

Overall, we showed that STING pathway activation does not give a major contribution to susceptibility to C-REV infection in human pancreatic cancer cell lines in vitro. These findings can be used to further improve the current state of oncolytic viral therapy. Our findings suggest that STING pathway activation in cancer cells is not critical for oncolytic viral therapy. Previous reports have shown that STING pathway activation in immune cells but not in cancer cells is critical for raising the cancer immune responses [41]. From these aspects, this paper can pave the way for studies evaluating the efficacy of combination therapies using oncolytic virus and STING activating drugs.

## Figures and Tables

**Figure 1 cells-10-01502-f001:**
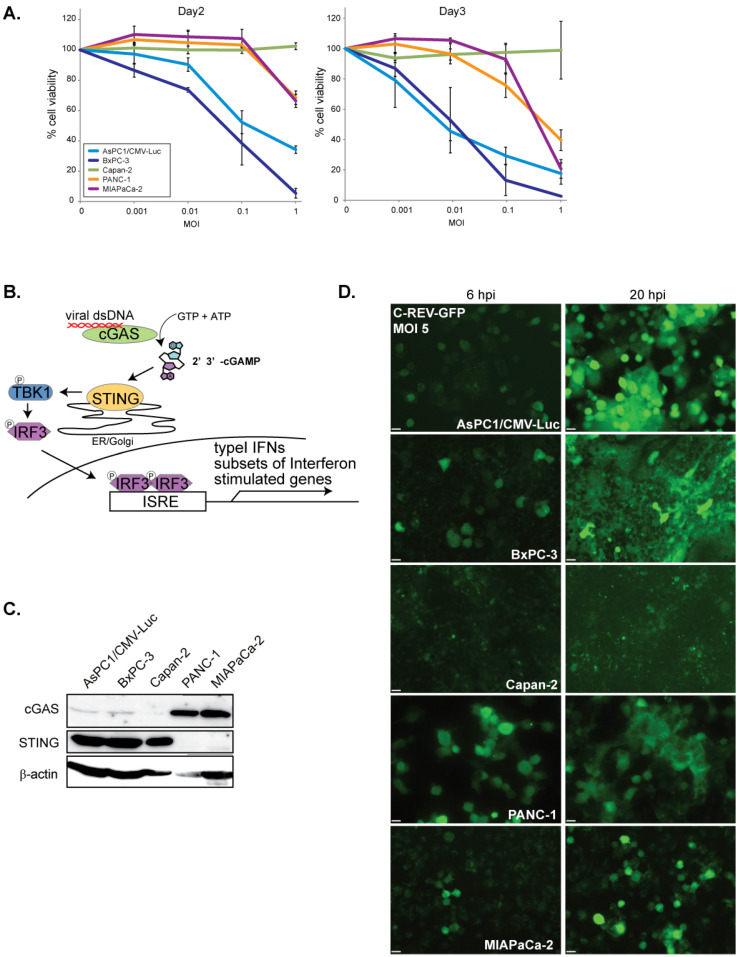
Susceptibility to C-REV on human pancreatic cancer cell lines. (**A**) Cytotoxicity of C-REV with several MOI against the indicated human pancreatic cancer cell lines on day 2 and day 3 by MTT assay. Data are presented as mean ± SEM of 3 independent experiments. (**B**) A brief illustration of the STING pathway. (**C**) Western blot analysis on STING and cGAS. (**D**) Images of the indicated cells infected with C-REV-GFP (MOI 5) at 6 hpi or 20 hpi. Scale bar represents 20 µm.

**Figure 2 cells-10-01502-f002:**
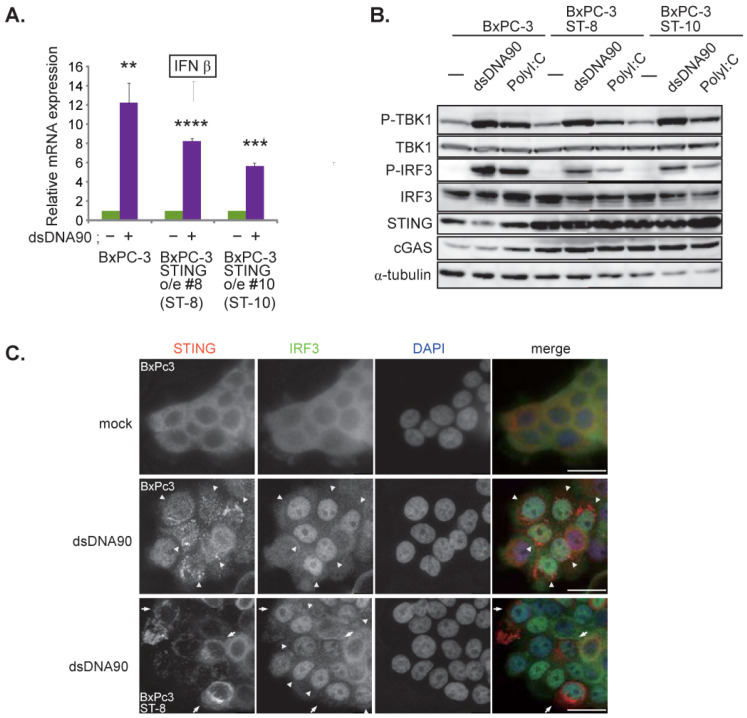
BxPC-3 cell line has a responsive STING pathway. (**A**) Induction of IFNβ. At 6 h after transfection of dsDNA90 (3 μg/mL) into BxPC-3, BxPC-3 ST-8 and BxPC-3 ST-10 cells, samples were analyzed by RT-qPCR. IFNβ expression was normalized relative to the expression of the G3PDH protein. Data are presented as mean ± SEM of 3 independent experiments. (**B**) Western blot analysis on the indicated proteins on the STING pathway. Cells were lysed after 6 h after transfection with dsDNA90 (3 μg/mL) or Poly (I:C) (3 μg/mL). (**C**) Images of the indicated cells at 6 h after transfection with dsDNA90 (3 μg/mL). Cells were stained with anti IRF3 antibodies and anti-STING antibodies. Arrowheads depict cells with stacked STING and with intranuclear translocation of IRF3. Arrows depict cells with over expression of STING. Scale bar represents 20 µm. ** *p* < 0.01, *** *p* < 0.001, **** *p* < 0.0001.

**Figure 3 cells-10-01502-f003:**
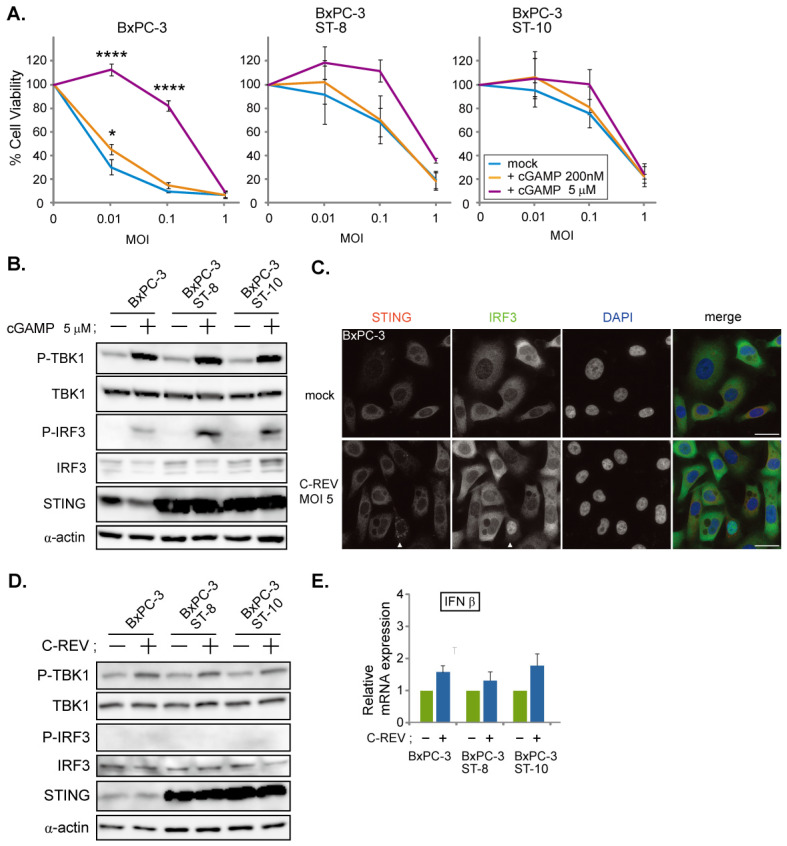
C-REV suppresses STING activation in BxPC-3 cell line. (**A**) Cytotoxicity of C-REV with indicated MOI against BxPC-3, BxPC-3 ST-8 and BxPC-3 ST-10 cells on day 3 by MTT assay. Cells were treated with digitonin and 2′3′-cGAMP (200 nM or 5 μM) for 1 h before viral infection. Data are presented as mean ± SEM of 3 independent experiments. (**B**) Western blot analysis on the indicated proteins on the STING pathway after treatment with digitonin and 2′3′-cGAMP (5 μM). (**C**) Images of BxPC-3 cells infected with C-REV (MOI 5) at 6 hpi. Cells were stained with anti IRF3 antibodies and anti-STING antibodies. Arrowheads depict cells with stacked STING and with intranuclear translocation of IRF3. Scale bar represents 20 µm. (**D**) Western blot analysis on the indicated proteins on the STING pathway. The indicated cells were lysed at 6 h after infection with C-REV (MOI 5). (**E**) Induction of IFNβ was analyzed by RT-qPCR on the indicated cells infected with C-REV (MOI 5) at 6 hpi. IFNβ expression was normalized relative to the expression of the G3PDH protein. Data are presented as mean ± SEM of 3 independent experiments. * *p* < 0.05, **** *p* < 0.0001.

**Figure 4 cells-10-01502-f004:**
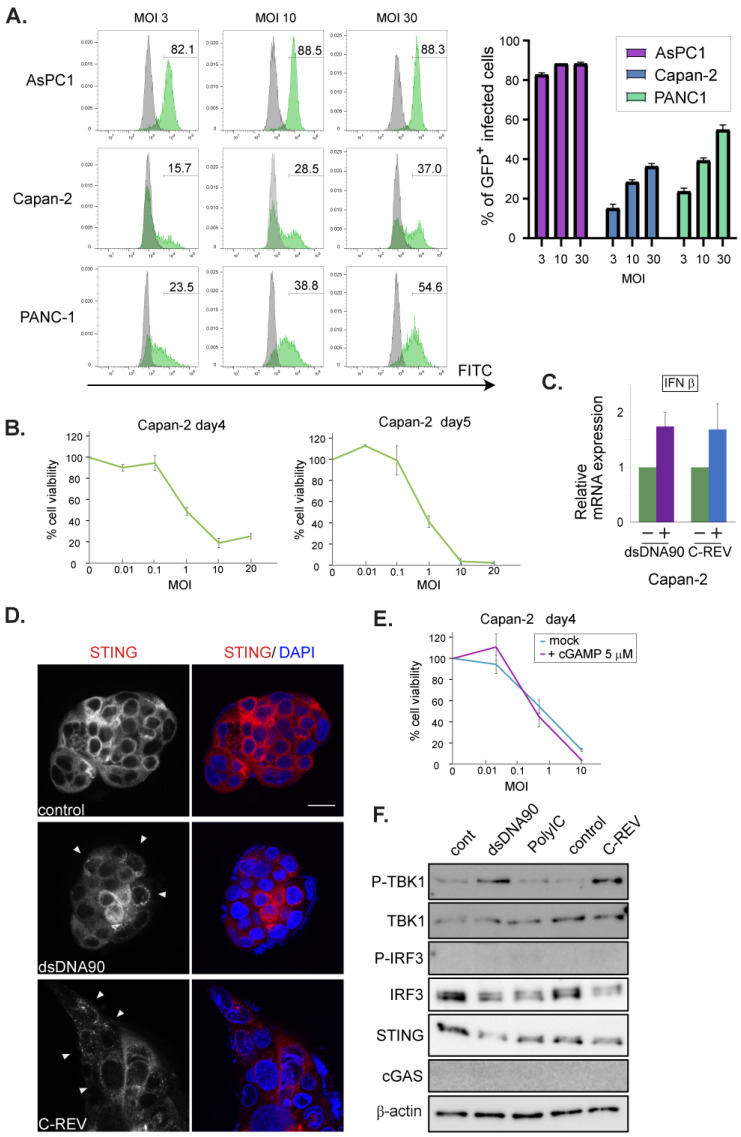
Low susceptibility to C-REV infection is not highly dependent on STING pathway activation in Capan-2 cell line. (**A**) Histograms of GFP+ infected cells. Respective cells were infected with C-REV-GFP at different MOIs (3, 10 and 30). After 12 h, cells were harvested and GFP was examined by flow cytometry. (right) Percentages of infected cells were calculated with the gate shown in left histograms. (**B**) Cytotoxicity of C-REV with several indicated MOI against Capan-2 cells on day 4 and day 5 by MTT assay. Data are presented as mean ± SEM of 3 independent experiments. (**C**) Induction of IFNβ. At 6 h after transfection of dsDNA90 (3 μg/mL), samples were analyzed by RT-qPCR. IFNβ expression was normalized relative to the expression of the G3PDH protein. Data are presented as mean ± SEM of 3 independent experiments. (**D**) Images of Capan-2 cells transfected with dsDNA90 (3 μg/mL), or cells infected with C-REV (MOI 5), after 6 h. Cells were stained with anti-STING antibodies (red). Arrowheads depict cells with stacked STING. Scale bar represents 20 µm. (**E**) Cytotoxicity of C-REV with indicated MOI against Capan-2 cells on day 4 by MTT assay. Cells were treated with digitonin and 2′3′-cGAMP (5 μM) for 1 h before viral infection. Data are presented as mean ± SEM of 3 independent experiments. (**F**) Western blot analysis on the indicated proteins on the STING pathway in Capan2 cells. Cells were lysed at 6 h after transfection with dsDNA90 (3 μg/mL), or Poly (I:C) (3 μg/mL), or lysed at 6 h after infection with C-REV (MOI 5).

**Figure 5 cells-10-01502-f005:**
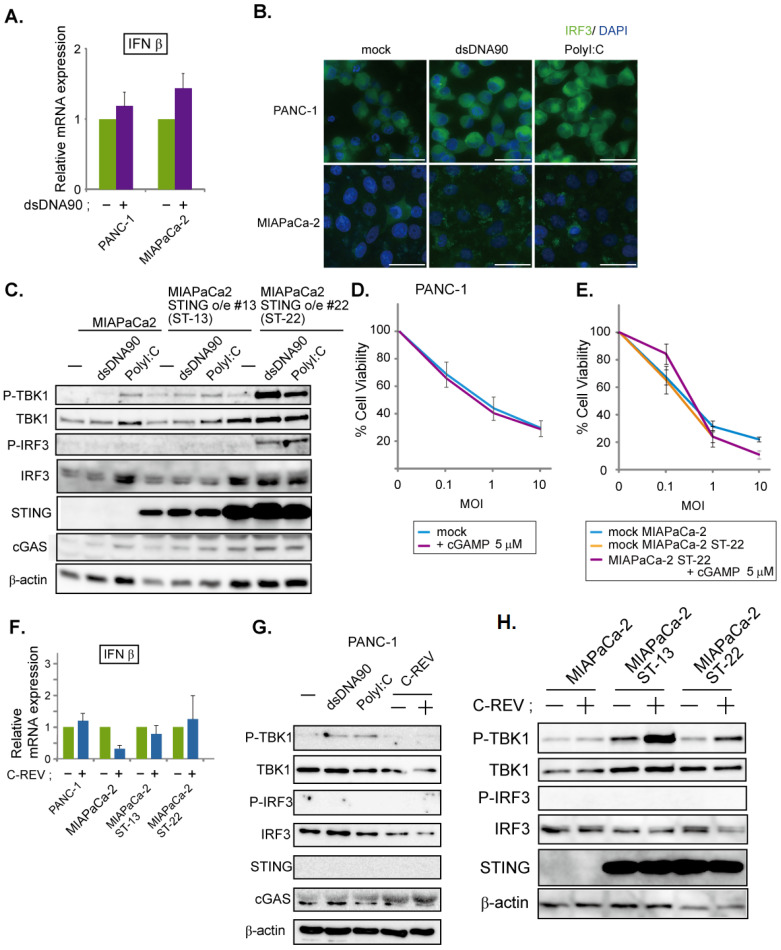
Low susceptibility to C-REV infection is not highly dependent on STING pathway activation. (**A**) Induction of IFNβ. At 6 h after transfection of dsDNA90 (3 μg/mL) into PANC-1 cells or MiaPaCa-2 cells, samples were analyzed by RT-qPCR. IFNβ expression was normalized relative to the expression of the G3PDH protein. Data are presented as mean ± SEM of 3 independent experiments. (**B**) Images of the indicated cells at 6 h after transfection with dsDNA90 (3 μg/mL), or Poly (I:C) (3 μg/mL). Cells were stained with anti IRF3 antibodies. (**C**) Western blot analysis on the indicated proteins on the STING pathway. Cells were lysed after 6 h after transfection with dsDNA90 (3 μg/mL), or Poly (I:C) (3 μg/mL). (**D**,**E**) Cytotoxicity of C-REV with indicated MOI against PANC-1 cells (**D**), MiaPaCa-2 cells or MiaPaCa-2 ST-22 cells (**E**) on day 3 by MTT assay. Cells were treated with digitonin and 2′3′-cGAMP (5 μM) for 1 h before viral infection. Data are presented as mean ± SEM of 3 independent experiments. (**F**) Induction of IFNβ was analyzed by RT-qPCR on the indicated cells infected with C-REV (MOI 5) at 6 hpi. IFNβ expression was normalized relative to the expression of the G3PDH protein. Data are presented as mean ± SEM of 3 independent experiments. (**G**,**H**) Western blot analysis on the indicated proteins on the STING pathway. Cells were lysed at 6 h after transfection with dsDNA90 (3 μg/mL), or Poly (I:C) (3 μg/mL), (**G**), or lysed at 6 h after infection with C-REV (MOI 5) (**G**,**H**).

## Data Availability

The data presented in this study are available on request from the corresponding author.

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
