# Peer review of "C-REV Retains High Infectivity Regardless of the Expression Levels of cGAS and STING in Cultured Pancreatic Cancer Cells"

_cells, 2021, doi:10.3390/cells10061502_

Round 1

Reviewer 1 Report

The eligibility and feasibility of oncolytic virus therapy as a novel therapeutic agent against pancreatic cancer are discussed as well as basic research for clinical trials, including a historical perspective and the current status of these novel agents. Even combination therapy, such as surgery with radiation and chemotherapy, has not significantly improved the survival rate of pancreatic cancer. Outcomes of pancreatic adenocarcinoma remain dismal despite extensive clinical investigation. Oncolytic viruses represent an emerging class of immunotherapeutic agents that have undergone extensive preclinical investigation and warrant further investigation in well-designed clinical trials. Therefore, new treatment modalities are highly needed.

The following point requires further attention:

  • Figure 1A presents susceptibility to C-REV on human pancreatic cancer cell lines 24 and 72 hrs post infections. Would be interesting to extend the kinetics study to 4/5 day pst infection in order to study the infectivity and replication of viruses.
  • Why the cell line Capan-2 (negative for cGAS) was less susceptible for the infectivity in comparison to AsPC... and BxPC-3 for instance (also negative for cGAS). 
  • Please correct typos and spelling errors throughout the manuscript (e.f lines: 250, 264...).
  • Did you check the expression of the INFalpha (IFN beta presented only in the paper).
  • What is the C-REV cell entry receptor, how it correlates with the expression of STING, cGAS? Is the expression level of the C-REV receptors cell entry affects the virus replication?
  • What is the adavantage of C-REV over T-VEC?
  • What is the study limitation?
  • How stated the findings contribute to the field of immuno-oncology and the use of oncolytic vectors?
  • Why authors did not perform immunological studies, e.g. with a presence of human immune cells in vitro; in vivo to confirm e.g. that advanced tumor cells become defective in the STING pathway to evade from the immune system and to further strengthen the data?
  • More studies need to be carried out to further confirm that cellular behaviors could be a determining factor for susceptibility in vitro as shown that STING pathway activation does not give a major contribution to susceptibility to C-REV infection in human pancreatic cancer cell lines in vitro.

Author Response

REVIEWER-1’s COMMENTS

The eligibility and feasibility of oncolytic virus therapy as a novel therapeutic agent against pancreatic cancer are discussed as well as basic research for clinical trials, including a historical perspective and the current status of these novel agents. Even combination therapy, such as surgery with radiation and chemotherapy, has not significantly improved the survival rate of pancreatic cancer. Outcomes of pancreatic adenocarcinoma remain dismal despite extensive clinical investigation. Oncolytic viruses represent an emerging class of immunotherapeutic agents that have undergone extensive preclinical investigation and warrant further investigation in well-designed clinical trials. Therefore, new treatment modalities are highly needed.

The following point requires further attention:

Comment-1: Figure 1A presents susceptibility to C-REV on human pancreatic cancer cell lines 24 and 72 hrs post infections. Would be interesting to extend the kinetics study to 4/5 day post infection in order to study the infectivity and replication of viruses.

Figure 1A which depicted our results of MTT assay at 48 hrs post infections (hpi) and 72 hpi suggested that more than 60 % of cells were already killed with MOI 1 in all the other cell lines except for Capan-2 cells. With this, we focused on Capan-2 cells. As suggested in the reviewer’s comment, we examined Capan-2 cells for MTT at 96 hpi and 120 hpi with broad range of MOI (MOI 0.01 – MOI 20) (new Figure 4B). To our surprise, Capan-2 cells showed sensitivity to C-REV. In relation to this new result, we also carefully confirmed the images of Capan-2 cells infected with C-REV GFP at 20 hpi (Figure 1D) and found that a few cells were GFP positive with faint signals (new Figure 1D). To further assess the infectivity of C-REV GFP on Capan-2 cells, we examined the GFP positive population by flow analysis. As expected, compared to highly susceptible cells such as AsPC1 cells in which more than 80 % of cells were GFP positive, there appeared only a partial GFP positive population in Capan-2 cells (new Figure 4A). These data indicate that Capan-2 cells are the least susceptible to C-REV among the 5 human PDAC cell lines.

Comment-2: Why the cell line Capan-2 (negative for cGAS) was less susceptible for the infectivity in comparison to AsPC... and BxPC-3 for instance (also negative for cGAS). 

As we mentioned in comment-1, new data gathered confirmed that Capan-2 cells were the least susceptible for C-REV infection. To elucidate the reason behind Capan-2’s low susceptibility, , we examined the STING pathway status of Capan-2 in response to dsDNA or C-REV infection (new Figure 4). As mentioned by the reviewer, indeed Capan-2 have similar expression patterns of STING and cGAS to AsPC1 and BxPC-3 howeverCapan-2 cells also harbored defects in activation of STING pathway and failed in induction of IFNβ (new Figure 4C). These results suggest that even with similar expression pattern of STING and cGAS, it is difficult to predict the status of STING pathway and susceptibility to C-REV. To confirm this, we also stimulated Capan-2 cells with cGAMP however this still failed to significantly increase the cytotoxic effect of C-REV infection on Capan-2 cells (new Figure 4E). Our western blotting analysis of Capan-2 cells also showed that although phosphorylation of TBK1 was detected, phosphorylation of IRF3 was not detected (new Figure 4F). Note that a similar pattern was observed fromPANC-1 cells and MiaPaCa-2 cells (Figure 5C and G). From these, we speculate that like in PANC-1 and MiaPaCa-2 cells, phosphorylation of IRF3 in Capan-2 cells might be dephosphorylated by protein phosphatase-1 which is reported as a negative regulator of IRF3. We discussed this point in new discussion part.

Comment-3: Please correct typos and spelling errors throughout the manuscript (e.f lines: 250, 264...).

Thank you for the comment. We rewrote typos and spelling errors as pointed out.

Comment-4: Did you check the expression of the INFalpha (IFN beta presented only in the paper).

Thank you for the suggestion. We agree that IFN alpha should be also assessed along with IFN beta. But since INF alpha has a broad family, it is difficult to assess the induction of all respective IFN alpha genes. Considering that different IFN alpha genes are reactive in the different cell lines and in the different conditions under which the cell lines are exposed to, we think that expression level of IFN alpha genes might not be a good tool in depicting the different activation status of the STING pathway among the five cell lines.  Moreover, elucidating which IFN alpha is reactive in each of the different cell lines under each of the different conditions would be extremely complicated. Lastly, most of the papers regarding STING pathways have shown that successful STING activation leads to the induction of IFN beta. With those careful considerations, we think that using IFN beta as a typical output for STING pathway activation is sufficient for our study.

Comment-5: What is the C-REV cell entry receptor, how it correlates with the expression of STING, cGAS? Is the expression level of the C-REV receptors cell entry affects the virus replication?

Thank you for the fundamental questions to improve our discussion. As C-REV is an HSV1 based oncolytic virus with natural deletion and mutation, the entry receptors for C-REV are basically the same as wildtype HSV1 which include Nectin1, HVEM, and 3-O-sulfated heparan sulfate. Despite this, to date, there is a lack of studies discussing in detail the nature and affinity of C-REV glycoproteins to known HSV-1 receptors. Additionally, despite searching extensively, we failed to find reports discussing the correlation between the expression level of those entry receptors and that of STING or cGAS. In general, if a cell is infected with multiple viral particles, that would arrow the virus to hijack the cellular system more rapidly and result in faster replication. Thus, higher expression level of entry receptors would increase the probability of successful viral infection, which would then lead to a rapid start of viral replication. In our new figure 4A, we conducted flow analysis on different cell lines after infection of C-REV-GFP with MOI 3, 10, 30. Looking closely at Capan-2 and PANC-1 in Figure 4A, we can notice that although the MOI, representing the number of viral particles, is increased tenfold, the percentage of infected GFP+ cells, which might indicate the hijacking efficiency of the virus, is increased only about two-fold. These results can hint that although increasing the number of viral particles might have a direct correlation with the probability of successful viral hijack, the expression level of entry receptors might be a more crucial defining factor in the probability of successful viral hijack since it increases the probability of successful viral entry.

Comment-6: What is the advantage of C-REV over T-VEC?

We appreciate the opportunity to discuss about this point. Although T-VEC is the only one oncolytic virus approved by FDA in US, T-VEC is an artificially genome modified virus, with deletions of ICP34.5 gene and ICP47 gene and insertion of human GM-CSF gene into the ICP34.5 gene locus. Since the ICP34.5 protein plays important roles against cellular host defense such as the STING pathway, it can be deduced that T-VEC is strongly attenuated. In addition, ICP47 protein has a role in inhibiting antigen presentation on MHC class-I for immune evasion. On the other hand, C-REV has an intact ICP34.5 gene and ICP47 gene but deletion of UL53 gene and LAT promoter region and possesses numerous mutations, which give enough attenuation to be considered an oncolytic virus. In addition to these deletions, C-REV also contains a duplicate of ICP27 gene which can inhibit phosphorylation of IRF3. Thus, we speculate that C-REV might have a more efficient inhibitory effect against STING pathway compared to T-VEC.

Comment-7: What is the study limitation?

This paper is mainly focused on cellular response against C-REV oncolytic virus in human pancreatic cancer cell lines in vitro. To better elucidate the role of the cellular defense system specifically the status of STING pathway activation on cancer cells against C-REV infection, we tried to separate the factors from the host’s immune system by utilizing only the in vitro system instead of the in-vivo system.

 Moreover, other reports have shown that in other cancer cell lines, an HSV1-based oncolytic virus has shown strong infectivity to the cells with defective STING pathway. Because this study only focused on five human pancreatic cancer cell lines and C-REV, it would be interesting to evaluate the effect of the intactness of the STING pathway against the infectivity of other HSV-1 based oncolytic virus in other cell lines in future studies.

Comment-8: How stated the findings contribute to the field of immuno-oncology and the use of oncolytic vectors?

We are hoping that our findings could support oncolytic viral therapy. Our findings suggest that STING pathway activation in cancer cells are not critical for oncolytic viral therapy. Previous reports have shown that STING pathway activation in immune cells but not in cancer cells is critical for raising the cancer immune responses. From those aspects, this paper can pave the way for studies evaluating the efficacy of combination therapies using oncolytic virus and STING activating drugs.

We also include these aspects into our new discussion part.

Comment-9: Why authors did not perform immunological studies, e.g. with a presence of human immune cells in vitro; in vivo to confirm e.g. that advanced tumor cells become defective in the STING pathway to evade from the immune system and to further strengthen the data?

Thank you for pointing this out. Because we focused only on the cellular response of STING pathway in cancer cells, we wanted to exclude immunological factors. Previous reports show that cGAMP can be incorporated into the viral particles and affect the responses in the neighbor cells and immune cells in the next round infection. Therefore, the level of cGAS activation can be a factor for activating of immune cells. This is interesting point to investigate in future experiments. We included this point into our new discussion part.

Comment-10: More studies need to be carried out to further confirm that cellular behaviors could be a determining factor for susceptibility in vitro as shown that STING pathway activation does not give a major contribution to susceptibility to C-REV infection in human pancreatic cancer cell lines in vitro.

We totally agree with the reviewer’s comment for this. Here we only suggest that STING pathway is not the major factor for determining susceptibility. As already pointed out, expression level of entry receptor for virus can be a factor. Also, as well known, the protein translation shut-off pathway can be a factor. Another factor would be from mutations behind the cancer cells including Kras dominant active mutation which is involved in the defective apoptotic pathway. We have to check those factors in future investigations.

Reviewer 2 Report

In this study the authors looked at the impact of defects in STING pathway in pancreatic cancer cell lines on the susceptibility to oncolytic virus C-REV infection.

The study used cell lines that are i) highly and ii) low susceptible to oncolytic virus infection and assessed the molecular signature of their STING pathway. The role of individual pathway effectors was investigated. The study concluded that differences in STING pathway functionality does not corelate with susceptibility to oncolytic virus infection.

The introduction is comprehensive; material and methods section offer enough detail and the results are clearly presented in logical subchapters. There are a few minor misspelling and text editing corrections:

Line 34: ”Pancreatic” should be lower case

Line 45: “1 4” ?

Line 57: “fibrolast”

Line 152: ‘’’Themedium’’

Line 192: ‘’p-value” change capital P-value at the beginning of a sentence.

Line 250: “whichshowed”

Line 253: golgi to be changed to capital G (Golgi)

Line 264: remove the “.” at the beginning of the paragraph

Line 276: “inlittle”

The discussion should be reviewed for style. There are several repetitions (“C-REV” in lines 375-76; “resistance” lines 384-85) and ambiguities (“stuff derived from infections” in line 365).

Please review Author Contributions for name accuracy (for example line 406: “abdelmoneim”)

Author Response

REVIEWER-2’s COMMENTS

In this study the authors looked at the impact of defects in STING pathway in pancreatic cancer cell lines on the susceptibility to oncolytic virus C-REV infection.

The study used cell lines that are i) highly and ii) low susceptible to oncolytic virus infection and assessed the molecular signature of their STING pathway. The role of individual pathway effectors was investigated. The study concluded that differences in STING pathway functionality does not corelate with susceptibility to oncolytic virus infection.

The introduction is comprehensive; material and methods section offer enough detail and the results are clearly presented in logical subchapters. There are a few minor misspelling and text editing corrections:

Comments:

Line 34: ”Pancreatic” should be lower case

Line 45: “1 4” ?

Line 57: “fibrolast”

Line 152: ‘’’Themedium’’

Line 192: ‘’p-value” change capital P-value at the beginning of a sentence.

Line 250: “whichshowed”

Line 253: golgi to be changed to capital G (Golgi)

Line 264: remove the “.” at the beginning of the paragraph

Line 276: “inlittle”

According to the reviewer’s comments, we carefully rewrote all the points.

Other comments:

The discussion should be reviewed for style. There are several repetitions (“C-REV” in lines 375-76; “resistance” lines 384-85) and ambiguities (“stuff derived from infections” in line 365).

Thank you for the comment on our discussion part. We carefully rewrote discussion part.

Please review Author Contributions for name accuracy (for example line 406: “abdelmoneim”)

Thank you for the comment. We carefully correct the part.

Round 2

Reviewer 1 Report

The authors provided satisfactory replies, edits, and improvements in the manuscript.